# Multi-Sensing Techniques with Ultrasound for Musculoskeletal Assessment: A Review

**DOI:** 10.3390/s22239232

**Published:** 2022-11-27

**Authors:** Jonathan de Oliveira, Mauren Abreu de Souza, Amauri Amorin Assef, Joaquim Miguel Maia

**Affiliations:** 1Graduate Program in Health Technology (PPGTS), Pontifical Catholic University of Paraná, Curitiba 80215-901, Brazil; 2Graduate Program in Electrical and Computer Engineering (CPGEI), Federal University of Technology of Paraná (UTFPR), Curitiba 80230-901, Brazil; 3Electronics Engineering Department (DAELN), Federal University of Technology of Paraná (UTFPR), Curitiba 80230-901, Brazil

**Keywords:** biosensing, ultrasound (US), electromyography (EMG), muscle-tendon unit (MTU), motor unit (MU), monitoring techniques, biomedical engineering

## Abstract

The study of muscle contractions generated by the muscle-tendon unit (MTU) plays a critical role in medical diagnoses, monitoring, rehabilitation, and functional assessments, including the potential for movement prediction modeling used for prosthetic control. Over the last decade, the use of combined traditional techniques to quantify information about the muscle condition that is correlated to neuromuscular electrical activation and the generation of muscle force and vibration has grown. The purpose of this review is to guide the reader to relevant works in different applications of ultrasound imaging in combination with other techniques for the characterization of biological signals. Several research groups have been using multi-sensing systems to carry out specific studies in the health area. We can divide these studies into two categories: human–machine interface (HMI), in which sensors are used to capture critical information to control computerized prostheses and/or robotic actuators, and physiological study, where sensors are used to investigate a hypothesis and/or a clinical diagnosis. In addition, the relevance, challenges, and expectations for future work are discussed.

## 1. Introduction

The musculoskeletal system is a complex structure made up of bones, muscles, and associated connective tissues that provides stability, movement, and support to the human body, allowing the performance of daily physical activities. It plays a fundamental role in respiratory mechanisms and helps maintain posture, body balance, and equilibrium [1]. However, several medical conditions can occur due to abnormalities in musculoskeletal function. In this framework, diseases such as myopathies, paralysis, and tremors, among others, are included. Even high-performance athletes or recreational sportsmen are susceptible to tendon and ligament problems, such as ruptures [2]. Therefore, understanding the physiologic aspects of the musculoskeletal system allows the assessment of the conditions that limit human functional performance and well-being [3]. This highlights the emphasis on body movement studies focusing on optimizing clinical diagnoses, such as identifying different states of relaxation, activity, and fatigue [4]. These studies also extend the evaluation of the dynamics, referring to the muscular vibrations of the motor stimulation units, to estimate the forces involved during the muscular contractions.

Since muscle strength in vivo cannot be measured directly, biomechanical computational modeling, based on the intrinsic mechanical and morphological properties of the muscle-tendon unit (MTU), represents an important tool for understanding the musculoskeletal system during locomotion [5,6,7]. This approach has been successfully applied and can describe the kinetic response of activated muscles, muscle force-producing characteristics, and individual muscle momentum at different angular contractions during motor tasks [8,9,10]. Hence, analysis of muscle contractions in the MTU plays a critical role in medical diagnoses and evaluations, monitoring, and rehabilitation in addition to assisting in movement prediction models used in prosthetic control [4,7,11].

Some researchers report the use of specific software to model, animate, and measure three-dimensional (3D) musculoskeletal figures, such as the Software for Interactive Musculoskeletal Modeling (SIMM) (MusculoGraphics Inc., Rohnert Park, CA, USA) [6]. However, one of the biggest challenges in simulating a musculoskeletal model is estimating the accuracy of MTU parameters on an individual-specific basis. According to Lemay and Crago [5], sensitivity analyses showed that the behavior of the musculoskeletal model tends to be more sensitive to values of MTU parameters. Thus, new approaches are expected to provide a better understanding and elucidation of muscle issues involved in lateral movement (expansion/retraction) of muscle fibers during locomotion [10,12].

In this sense, the inclusion of additional techniques, such as ultrasonography, have been widely applied to investigate changes in the morphological structure of the tissue [13]. These modalities are based on ultrasound (US), in which acoustic waves are emitted, typically in the order of MHz, which propagate and interact with the investigated medium [14,15,16]. Currently, US systems are used to assist clinical investigations and diagnoses, patient monitoring, and rehabilitation, contributing to the generation of images of internal anatomical structures and blood flow in different modes, namely as A-mode (amplitude), B-mode (brightness), M-mode (motion), continuous-wave (CW) Doppler, pulsed-wave (PW) Doppler, and most recently, elastography mode [17]. 

It is worth mentioning that most commercial ultrasound systems have a typically “closed” architecture, making flexibility for testing and research difficult. On the other hand, there are few open research systems such as the OPEN System Phased array device (Lecoeur Electronique, Chuelles, France) [18], the Ultrasonix 500RP (Ultrasonix Medical Corp., Richmond, BC, Canada) [19], and the Vantage Research System (Verasonics Inc., Kirkland, WA, USA) [20]. Additionally, some research groups developed their own platforms, such as the Ultrasound Advanced Open Platform (ULA-OP) 256 [21], the Remotely Accessible Software Configurable Multichannel Ultrasound Sampling (RASMUS) [22], and the Synthetic Aperture Real-Time Ultrasound System (SARUS) [16], as shown in Figure 1.

However, regarding ultrasound (US), there is a lack of investigation towards US data fusion for assessing musculoskeletal system. To illustrate, here are only few examples: in the review paper from Grushko et al. [23], there was an extensive review for biosensing approaches focused on prosthetic control in trans-radial prostheses electromyography (EMG), electrical impedance tomography (EIT), near infrared spectroscopy (NIRS), sonomyography (SMG), force myography (FMG), and phonomyography (PMG). Another similar review presented by Zheng [24] covered FMG, EMG, and EIT as applied to human–machine interface (HMI), i.e., for medical applications, but did not include US data. Therefore, to cover this gap in the literature, including different and complementary modalities, this paper presents some multi-sensing and hybrid techniques, which covers most of the applications, and it has been a vital piece for biomedical engineering. Then, this review paper provides a guide to recent works, studies, and experiments within biomedical signals also combined with US. Additionally, to elucidate, this paper compiles feasibility, performances, and results regarding different applications such as musculoskeletal assessment in HMI devices, prosthetic control, the healthcare field, and physiological studies. 

This paper is divided into the following sections: Section 2 presents the background about the techniques to be explored in this review; Section 3 covers the researchers analyzed here; Section 4 presents some suggestions and comments by the authors about new perspectives in the area; and Section 5 summarizes this review. 

## 2. Biosensing Techniques

In the last decade, the use of combined traditional techniques has grown to assist in the monitoring and quantification of lateral oscillations of muscle activity present in the isometric contraction movement performed by the skeletal muscles [7,8,10]. Typically, these methods quantify information about muscle condition that is correlated with neuromuscular electrical activation and the generation of muscle force and vibration [25]. Consequently, its temporal and spectral analysis can help in the determination of muscle fatigue [26]. This section presents some biosensing techniques that guided this work, which are also illustrated in Figure 2.

### 2.1. Sonomyography (SMG)

The US imaging modality includes sonomyography (SMG), which uses transducers with piezoelectric properties that make it possible to associate the “sound” of muscle vibrating waves with muscle contraction, being more sensitive to pressure and low-frequency vibration [27]. Ultrasonic waves can penetrate a few centimeters below the skin and are reflected in a way that returns both superficial and deep information [28,29]. This reveals its application in the observation of changes in muscle thickness, cross-sectional area, fascicle length, and pennation angle (PA) during a muscle contraction [30,31,32]. These characteristics may provide enough information for prosthetic control [13]. The device used to capture this signal is a probe, which needs to be fixed in a position to perform the scan to acquire the image [33]. On the other hand, this limitation makes its application difficult in terms of dynamic conditions.

### 2.2. Elastography 

In addition to B-mode US imaging, there is a relatively recent modality to assess tissue stiffness, classified as elastography [34]. Shear wave elastography (SWE) allows the quantitative characterization of mechanical properties of tissues and has shown great potential in numerous applications [35]. In the last decade, the 2D-SWE has been used for clinical evaluation of the breast [36,37,38], liver [39,40,41], thyroid [42,43], and prostate [44,45]. Recently, SWE has also been investigated for application in musculoskeletal tissues, focusing on detecting elasticity during contraction and relaxation in vitro and in vivo [46,47,48,49,50]. Techniques focused on present elastography possibilities such as 3D tracking using matrix transducers with high accuracy rates, allowing the analysis of parameters such as response time, shear wave velocity, and analysis of regions of interest in real time in different tissue layers [51,52].

### 2.3. Mecanomyography (MMG)

The mechanomyography (MMG) technique has been used to evaluate the mechanical response at low frequency (2–200 Hz) [53], that is, a vibration that propagates through the skin surface [54], and the lateral oscillation is caused by the contraction of muscle fibers. However, the fact that there are no well-established sensors on the market and problems due to acoustic interference makes the technique more restricted [25].

### 2.4. Electromyography (EMG)

A well-explored technique to monitor muscle activity is surface electromyography (sEMG), which consists of identifying the electrical contribution made by an active motor unit (MU) during the contraction of muscle fibers [27]. Even though it is a popular and widely used technique [4], it presents some difficulties when measuring deeper muscle activities due to skin impedance, high sensitivity regarding sensor positioning, signal attenuation, low signal-to-noise ratio (SNR), and crosstalk [8,55,56].

### 2.5. Additional Techniques

There are some more modern non-invasive techniques for assessing muscle activity, such as electrical impedance tomography (EIT), which uses several surface electrodes tied around the user’s body to measure tissue impedance in the section plane [23,57], and near infrared spectroscopy (NIRS), which allows monitoring of muscle perfusion and oxygenation during contractions from light emitting diodes (LED) that emit NIR light into the tissues, while photodetectors measure the amount of light scattered close to the tissue [23,58].

Some additional information about the muscle can be acquired through goniometer to measure the angle of the joint [59] and plates attached to the floor to measure the reaction force with the ground [60]. Pressure sensors and dynamometers can be used to identify the movement and strength of the muscle joint [32,61] in conjunction with infrared thermography and motion capture with cameras and reflective markers to measure kinetics and kinematics [60].

To provide a better understanding of the muscle issues involved in a muscle contraction in the musculoskeletal unit, there is a demand for further investigations using the joint approach of EMG, MMG, force sensors, goniometer, and SMG with B-mode and elastography images. Even wearable and compact systems for investigating the forces involved in the dynamics of muscle vibrations are of great interest even for future developments for dynamic applications [10,12].

## 3. Research and Application

Several research groups have been using multi-sensing systems to carry out specific studies in the healthcare field to investigate deeper aspects of the musculoskeletal system. Even though there are vast applications for US in this topic, we divided the modalities of each study to standardize the whole work and ease the task for the reader. First, we bring the applications of US into HMI, in which sensors are used to capture critical information for controlling robotic prostheses and/or actuators. Later, it comes the physiological studies, where sensors are used to investigate a hypothesis, clinical diagnosis, and/or mathematical models.

### 3.1. Human–Machine Interface

The extraction of features regarding the morphological changes of the muscle can be done through US scanning in A-mode, B-mode, or even M-mode. Figure 3 presents wearables embedded with US transducers and sEMG sensors for prosthetic applications. Table 1 shows the specific characteristics and equipment used in each work reported in this investigation. 

#### 3.1.1. A-Mode Ultrasound

In this modality, the signal provides information about echo amplitude, implying in 1D display, pointing only in the direction where the probe is facing and resulting in waveforms with spikes and peaks at the interface of two different tissues [64].

In the work of Guo et al. [13], the performance of the surface EMG and 1D SMG signal was investigated from three different wrist extension movements, and changes in tissue thickness were analyzed in real time. This same group also published a work in 2011 [8] that presented a study comparing the strength and angle control performance of 1D SMG signals and sEMG during an isometric contraction and wrist extension. The results indicate that the SMG signal can generate a more consistent prosthetic control performance compared to the EMG signal in isometric contraction and pulse extension activities.

A portable sEMG and A-mode US hybrid sensing system for human–machine interface (HMI) was developed in Xia et al. [62], which includes a solution with an armband design, a printed circuit board hardware, and a multi-source procurement strategy. They also presented the feasibility of implementing a hybrid sEMG and A-mode US proposal. It also points out that this combination adds up when investigating muscle contraction at a greater depth in the muscle (US > sEMG) and, at the same time, brings a more concise reading when the hand is in a resting state (sEMG > US). However, there is still room for wireless communication to improve the system compactness.

HMI was also implemented in a US imaging study conducted by Yang et al. [56]. For this experiment, the muscle thickness was extracted from a US image to perform a non-linear mapping between the muscle thickness and the degree of fatigue, concluding that there was a linear relationship between muscle strain and normalized biceps brachial torque. In this context, a major issue to address is the transducer shift, which occurs more often during wrist pronation and supination.

Recently, a group developed [65] a wearable system with 8 A-Mode US channels and miniaturized with US transducers for sensing muscle morphological deformation during limb movement via multiple perspectives of muscle contraction monitoring. In addition, it can be applied for prosthesis control, virtual reality interaction, and human movement analysis, among other applications. Yet, there is a need for deep investigation about the number of transducers to achieve optimal results for prosthesis control. In the same topic, the group from Lu et al. [66] proposed a scheme based on wearable A-Mode Ultrasound for gesture recognition. In addition, they contributed to real-time experiments by evaluating four online performance metrics, which achieved high performance. Furthermore, they also compared different features in 3D map and introduced relative offset rate of quantitative analysis. Furthermore, it was shown that A-Mode ultrasound has some limitations, capable of detecting depth in only 1D, which means that offset of the transducers and different arm posture can decrease its recognition accuracy. 

#### 3.1.2. B-Mode Ultrasound

This type of scan can produce 2D images of the underlying tissues [64], also called the brightness mode (B-Mode). In this modality, measuring the distance between two fascias becomes easier, as it presents visualization of a cross-section of anatomical structures [67].

The group [68] proposed a new prosthetic control model by classifying the movement intention and making it more intuitive based on the control’s proportional position without multiple degrees of freedom (DOF). They also employed a portable US system in which the participants performed predefined hand movements while data from the forearm were captured. For classification, the k-nearest-neighbor (kNN) method was used. It has been shown that both healthy subjects and amputees can use the SMG signal to control the 1-DOF virtual cursor position simply by flexing the forearm muscles appropriately. The SMG offers position control capability in a single DOF. They also used a sonomyography (SMG) application although this approach is still too robust for commercial prosthesis. 

Wang et al. [63] performed another experiment where B-mode US signals were tried and compared with sEMG signals to recognize movement intention in people with trans-radial amputation. The results show that US signals can achieve better accuracy than surface electromyography signals in specific and controlled test environments; however, it ends up being more sensitive to artifacts. 

Jahanandish et al. [11] tried to assess the feasibility of a reliable prediction of kinematics arising from joint movements by using as a basis the characteristics extracted from US in lower-limb muscles. Additionally, the amount of time that US precedes joint action during experimental weightless knee flexion/extension was characterized. 

The same group also evaluated the wearable US imaging (WUSI) as a new modality for continuous classification of five discrete modes of ambulation [69]: level, incline, decline, ascending and descending ambulation ladder, and performance of reference concerning the EMG. They concluded that the wearable US sensing significantly improves the classification accuracy of multiple ambulation modes compared to sEMG. It was also found that US sensing can be performed with fewer sensors spread over the limbs compared to other sensing modalities.

The neuromuscular signal can be used to predict moments such as dorsiflexion, which is generated by the tibialis anterior (TA) muscle. Therefore, the group [70] tried to compare some approaches to this signal by analyzing their prediction performance. Surface electromyography signals and three signals derived from US imaging were used: fiber length, pennation angle, and echogenicity. As a conclusion, they observed an increase in the dorsiflexion moment prediction performance when using the pennation angle, fascicle length, and sEMG variables compared to the echogenicity variable.

#### 3.1.3. M-Mode Ultrasound

This modality is called motion mode (M-Mode), which reflects motion aspects of connective tissues within muscles by using a narrow beam to produce a 1D view of anatomical structures over time [67,71,72].

This mode was tried in the work of [73], in which they tested its feasibility for finger and wrist movement recognition by analyzing 13 different gestures. The separability index (SI) and the semi-major axis mean (MSA) were used, which were calculated to quantify the spatial characteristics of both modalities. They found that for the M-mode, there was a higher value for the SI and a lower value for the MSA compared to B-mode. In the classification for motion recognition, the M-mode presented a slightly lower result in only two gestures, being slightly superior in the others. In general, the performance of both modes was not significantly different even though in this experiment, the M-mode presented less spatial information. They believed that this result was due to the high sampling rate of M-mode when compared to B-mode, thus compensating for the smaller amount of spatial information. This implies a potential in the study of M-mode to build an HMI with less delay and better performance.

Still, in this modality, it would be interesting to investigate whether the M-mode has the same sensitivity regarding the fixation of the US probe in relation to the B-mode since it has a high sampling rate.

#### 3.1.4. Multi-Sensor Studies without Ultrasound

In the study by Geng et al. [74], the classification of EMG and MMG signals through the type of movement and position with multiple classes, performed in different limb positions of subjects with trans-radial amputation, was analyzed and compared.

**Table 1 sensors-22-09232-t001:** Summary of HMI Research.

Authors	Signals	Region	Specifications	Data and Features	Subjects
Guo et al., 2009 [13]	US, EMG	Forearm	10 MHz A-Mode Transducer, with a diameter of 6 mm inserted into a support made with silicone gel of 20 mm in diameter; A-mode with 17 Hz frame rate	Muscle Deformation Signal (SMG); RMS EMG	16 (8 men and 8 women)
Guo et al., 2011 [8]	US, EMG, Force and Direction Sensors, Goniometer	Long Radial Extensor of the Carpus	10 MHz A-Mode Transducer, 7 mm diameter	RMS EMG; 1D SMG; Wrist Angle; Force	16 (8 men and 8 women)
Geng et al., 2012 [74]	EMG, MMG	Forearm	Classifier: LDA	EMG: MAV, ZC, WL, and SSC. MMG: MAV, Variation, and Maximum Value	5 (4 men and 1 woman)
Yang et al., 2018 [56]	US	Forearm	5 MHz A-Mode Transducer, 14 mm diameter and 18 mm height; Classifiers: LDA and SVM	US A-Mode, Method for Feature Extraction: Segmentation and Linear Fitting	8 men
Xia et al., 2019 [62]	US, sEMG	Forearm	5 MHz Linear US Transducer; Custom EMG and US Acquisition Module.	EMG: MAV, WL, ZC, SSC, and AR6. US: MSD	8 men
Dhawan et al., 2019 [68]	US	Forearm	US images at 15 Frames per Second (FPS). Classifier: kNN	Position Error, Error Stability, Task Completion, and Movement Time	5 (4 unilateral amputations and 1 bilateral upper-limb amputation) + 5 healthy (control group)
Wang et al., 2020 [63]	US, EMG	Forearm	5 MHz US Transducer with 10 FPS	EMG: RMS, MAV, WL, and AR4	One subject with trans-radial amputation
Botros et al., 2020 [75]	EMG	Wrist, Forearm	Classifier: LDA and SVM	EMG: RMS, MAV, WL, ZC, and SSC	21 subjects (14 men and 7 women)
Jahanandish et al., 2020 [11]	US	Rectus Femoris (RF)	US Transducer with 50 dB Dynamic Range	Muscle Thickness; Angle between Aponeuroses; Pennation Angle; Fasciculus Length; echogenicity	9 (5 men and 4 women)
Zhang et al., 2021 [70]	US, EMG	Ankle, Tibialis Anterior (TA)	Linear US Transducer with 6.4 MHz Center Frequency. Models: LR, FFNN, and HNM	Pennation Angle, Fasciculus Length, Echogenicity, sEMG Mean RMS, Maximum Force, and Moment	3 men
Souza et al., 2021 [76]	EMG, ACC, Sliding and Force Sensors	Forearm	12 EMG and 36 ACC. Classifier: PCA, LDA, RFT, and MLP	EMG: MAV, RMS, WL, logRMS, Variance, Kurtosis, and Skewness	Dataset NinaPro DB2: 34 (12 women and 28 men); DB3: 11 trans-radial amputee (11 men)
Rabe et al., 2021 [69]	US, EMG	Rectus Femoris (RF), Vastus Medialis (VM), Vasto Intermediate (VI)	128-element Linear US Transducer with a Transmission Frequency of 7.5 MHz and a Dynamic Range of 50 dB. Custom Synchronization Software Displays US Images and EMG Signals with 1 ms Temporal Resolution.	5-EMG, 8-EMG: MAV, SSC, ZC, WL, and 2 coefs. AR4 US: Aponeurosis Angle, Muscle Thickness, Fascicle Length, and Echogenicity	10
X. Yang et al., 2021 [65]	US	Wrist	A-Mode Transducer with a Diameter of 9 mm and Height of 11 mm	Wrist Rotation Angle (Mean, Standard Deviation, Maximum, Minimum, Sum, Skewness, Kurtosis)	8 men
J. Li et al., 2022 [73]	US	Forearm	US 5–12 MHz Linear Transducer with 50 × 4 mm Contact Surface; Videos Stored at 30 Hz. Classifier: SVM and BP	Separability Index (SI) and Mean Semi-Principal Axis (MSA)	8 (7 men and one woman)
LU et al., 2022 [66]	US	Forearm	4-channel A-Mode Ultrasound at 10 FPS. Classifier: LDA, SVM, and Naive Bayes (NB)	Mean, Variance, and Energy	10 (8 men and 2 women)

AR, Autoregressive Model; BP, Back Propagation; FFNN, Feed-Forward Neural Network; HNM, Hill-type Neuromuscular Model; kNN, k-Nearest Neighbor; LDA, Linear Discriminant Analysis; LR, Linear Regression; MAV, Mean Absolute Value; MLP, Multilayer Perceptron; NB, Naive Bayes; PCA, Principal Component Analysis; RFT, Random Forest; RMS, Root Mean Square; SSC, Slop Sign Change; SVM, Support Vector Machine; WL, Waveform Length; ZC, Zero Crossing.

In Botros et al. [75], a systematic review was performed on the applicability of hand gesture recognition from EMG signals acquired at the wrist. As a result, it was found that the acquisition of signals from the wrist suffers fewer losses from noisy artifacts than signals obtained from the forearm.

In the research of Souza et al. [76], a real-time hardware and software control of an open system hand prosthesis was performed using EMG signal, slip sensors, and force sensors in instrumentation. In this case, machine learning was applied for gesture recognition. The experimental result shows that combining EMG and accelerometry (ACC) data improves recognition accuracy. It was also found that the body mass index (BMI) of each volunteer affects the classification accuracy, so the lower the BMI, the better the accuracy.

### 3.2. Physiological Studies

As shown before, US data can provide useful knowledge at the muscle tissue level, such as thickness change (TC), cross-section area (CSA), echogenicity, joint angle, pennation angle, and fascicle length [67]. These data also have good correlation with other techniques, such as EMG, MMG, muscle force, deformation, joint velocity [27]. For EMG and MMG signals, RMS and MPF features are useful to analyze muscle activation and movement tracking during an exercise or a physical activity. Even though EMG is an electrical muscle activity, it also shows information at neuromuscular junction level [54], as for the mechanical signal, MMG can illustrate sound waves at the muscle level [53].

In addition, US modalities are vastly deployed with other biomedical signals to support clinical evidence [10,27] to enhance accuracy in detecting irregularities, muscle dystrophy and myositis [77], and neuromuscular disorders using electromechanical (EMD) indicator to predict, e.g., spastic cerebral palsy and ligament laxity [54]. It is applied with the golden-standard EMG/neuro conduction study (NCS) to improve false-negative results (10–25%) in carpal tunnel syndrome (CTS) and nerve damage assessment [78,79], enabling real-time dynamic visualization of moving tissue, blood flow with Doppler imaging, and serving as guidance throughout the procedure, protecting the target and surrounding soft tissue structures in diagnosis and therapeutic needle EMG [78]. Quantitative muscle ultrasound (QMUS), which uses the mean echogenicity or calibrated backscatter technique, is currently one of the most reliable techniques to assess classic neuromuscular disorders, e.g., Duchenne muscular dystrophy (DMD), Pompe disease, or spinal muscular atrophy [77]. Figure 4 shows the experimental sensors placements and setups for specific movement studies, where (a) and (b) work with dynamic muscle behavior and (c) with reflective markers to track the joint angle. This approach helps in building a more robust and precise analysis and correlations. Furthermore, it could also be helpful in mathematical models. Table 2 contains information about setups and the systems used for each experiment.

**Figure 4 sensors-22-09232-f004:**
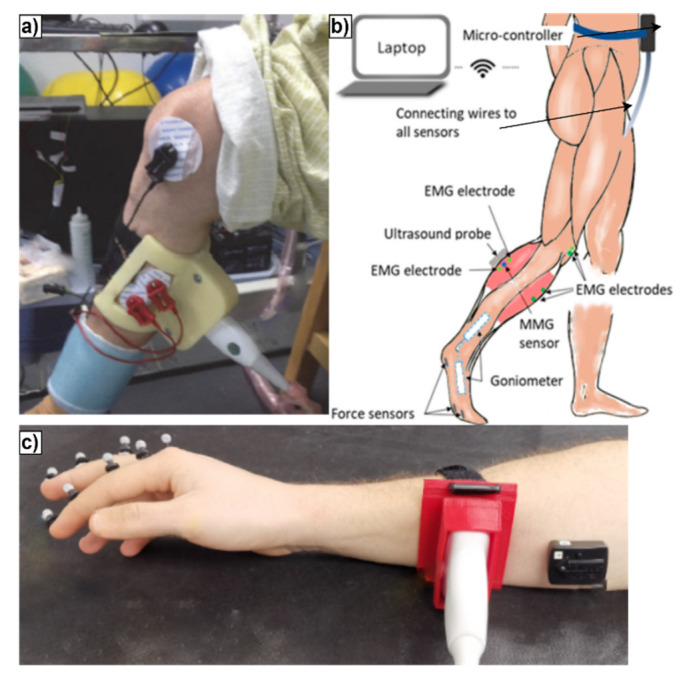
Adapted photographs of multi-sensor setups for physiology research: (**a**) experimental setup done by [80] with EMG, goniometer and US; (**b**) wearable mobile sensing system with real-time US imaging, EMG electrode, mechanomyography electrode, force sensors and goniometers placed in the shank and foot, developed by [10]; and (**c**) experimental setup with reflective markers placed in the back side of the hand, with US probe supported by an armband along the forearm and sEMG electrodes, presented by [81].

#### 3.2.1. Signal Correlations

A system was used for simultaneous EMG, MMG, and SMG collections in [27] to study the rectus femoris (RF) muscle during an isometric contraction ramp for different contraction speeds. Then, an algorithm for monitoring the US image was developed to extract the cross-sectional area automatically. This study demonstrated that the SMG–torque relationship is less affected by contraction speeds, which differs from the EMG and MMG signals during an isometric ramp contraction, pointing out that the SMG can provide important morphological parameters of a continuous contraction.

In study of [82], EMG and SMG were captured separately in several series of plantar-flexion movements to assess the TA muscle of subjects with hemiplegia. They traced the relationship between TA muscle thickness changes and the patients’ muscle strength, which showed that SMG may be a promising option for quantitative estimation of muscle strength level (MSL) for patients with hemiplegia during rehabilitation. Yet, there is a need for further investigations with a larger number of subjects clustered according to age, gender, or pathological approaches.

Through analytical cross-correlation, the group [83] compared the thickness of the gluteus muscle of female subjects derived from the US imaging in relation to kinematics, kinetics, and sEMG. They observed that hip muscle thickness, sEMG signal, kinematics, and kinetics are related in a time spectrum. The authors also showed that the electromechanical delay that occurs is approximately 99 ms in the morphological responses of the US imaging. As these findings were achieved with a small female group, it could not be generalizable for a broader population or for males.

Without the ultrasound signal, to present a low-cost sensing system, the group [4] a developed a system capable of monitoring muscle activity in the face of physical activity. The objective of the study was to segment the phases of muscular activity and compare the results obtained from RF muscle with MMG, EMG, and inertial measurement unit (IMU) during the exercise. The result demonstrated how the combination of information from a muscle contraction, provided by the EMG and MMG signal, and from a dynamic movement acquired from the IMU can improve the understanding of human muscle and activity movement in a specific period of contraction.

#### 3.2.2. Gait Analysis

The dynamic behavior of the muscle during a gait of a patient after hemiparetic stroke was investigated by Chen et al. [80]. They aimed to examine the contraction patterns of muscle activity in the lower extremity and compare the affected side with the unaffected one. An image-tracking algorithm was developed to automatically extract continuous changes from the gastrocnemius muscle, making it possible to identify muscle morphology at high temporal resolution.

Another group [10] developed a mobile sensing system with real-time US imaging to capture and analyze human movement by applying a gait analysis. Their approach was rated with moderate-good reliability across the various channels. Some changes were identified in the lateral head of the gastrocnemius (GM) muscle during a gait through the comprehensive assessment of muscle activity by multiple modalities integrated into the system. One limitation of this type of study is the small size population. As the developed system is a prototype, its frame rate reaches only up to 10 Hz, and there is still room for improvements, especially regarding the wired connections.

B-mode US imaging was also used to measure the in vivo contractile dynamics of the soleus (SO) muscle fascicle during an exoskeleton-assisted walk [12]. Rotational stiffness of the exoskeleton has been shown to alter the tuned catapult behavior of the ankle plantar flexor. As stiffness increased, fascicle length and velocity increased, and they became more likely to be affected by the force-producing capacity of the muscle as an effect of muscle strength changes by activation calculation. 

#### 3.2.3. Ultrasound in Motor Unit Analysis

Structural parameters of the muscle were studied through US images on the effect of neuromuscular electrical stimulation (NMES) by Qiu et al. [84]. The authors found that the SMG can be an effective alternative for detecting and pacing current response in an NMES application.

A new framework was developed by Zheng et al. [81] to estimate muscle movement in a transverse plane and to detect the activated muscle in a transverse US image. An algorithm was used to trace the displacement of the muscles and generate a strain field in a transverse plane. The first experiment found that the developed method can identify muscle contractions in both superficial and internal layers from intramuscular acquisitions. In the second experiment, the technique was insensitive only for passive shortening of the muscle since the speed joint movement was small. There was also no activity in the stretch reflex, causing the muscle strain to be minor in this condition.

**Table 2 sensors-22-09232-t002:** Summary of Physiological Study.

Authors	Signals	Region	Specifications	Data & Features	Subjects
Chen et al., 2012 [27]	US, EMG, MMG	Rectus Femoris (RF)	NI PCI with 25 FPS and 0.15 mm resolution; Image Processing Method via “Deformation Tracking” for Continuous CSA Extraction	RMS EMG, RMS MMG, Cross-Section Area (CSA), and Torque	9 (6 men and 3 women)
H. Li et al., 2014 [82]	US, EMG	Tibialis Anterior (TA)	US Transducer with 7.5 MHz at a Detection Depth of 70 mm; 128 frames in 10 s	SMG: TC (Thickness Change). EMG: RMS	12 (9 men and 3 women)
Qiu et al., 2016 [84]	US, NMES	Quadriceps Femoris (Rectus Femoris (RF), Vastus Intermediate (VI), Vastus Medialis (VM), and Vastus Lateralis (VL))	US 3.5 MHz Transducer; Capture Card for 25 FPS B-Mode; Stimulation Pulse of 300 US and Frequency of 25 Hz, Stimulation Current from 0 to 150 mA (470 Ohms)	Muscle Thickness and Joint Angle (JA)	7 (5 men and 2 women)
Chen et al., 2017 [80]	US, EMG, Goniometer	Gastrocnemius (GM)	Video Capture Card with 25 FPS	RMS EMG, Pennation Angle (PA), and Joint Angle (JA)	12 (9 men and 3 women) patients with chronic and subacute stroke
Ma et al., 2019 [10]	US, EMG, MMG, Force Sensor, Goniometer	Lateral Head of Gastrocnemius, Tibialis Anterior (TA), Ankle, Heel, Forefoot	US Probe with 7.5 MHz; VICON with 250 Hz; Power Plate with 1 kHz; Ultrasound Images at 10 FPS	Normalized EMG and MMG peak value; CSA; Joint Angle and Force	10 (7 men and 3 women)
Woodward et al., 2019 [4]	EMG, MMG, IMU	Rectus Femoris (RF)	IMU with Triaxial 2000° per Second Gyroscope (STMicroelectronics L3G4200D), 16 g Triaxial Accelerometer (Analog Devices ADXL345), 8 G Magnetometer (Honeywell HMC5883L), and a −500 to +9000 m Barometer (Bosch BMP085)	MPF and RMS EMG and MMG signals	5 (4 men and 1 woman)
Ling et al., 2020 [54]	US, EMG, MMG, Force Sensor	Tibialis Anterior (TA)	Ultrasound images at 20,000 FPS with an imaging depth of 3.5 cm	Movement Onset Time: EMG, MMG, SMMG, and Force	P1: 7 (3 men and 4 women); P2: 8 (5 men and 3 women)
Nuckols et al., 2020 [12]	US, EMG, Force Sensor, Calorimetry System, Motion Capture System	Soleus (SO)	7.5 MHz 96-Element US Transducer: Automatic Software to Determine FL and PA; Vicon with 44 Reflective Markers to Capture 6 DOF of the Foot, Shin, Thigh, and Pelvis	Joint Velocity, Joint Angle, RMS sEMG, Muscle Force, Fasciculus Length (FL), CSA	11 (7 men and 4 women)
DeJong et al., 2020 [83]	US, EMG	Gluteus	8 MHz US Wireless Linear Transducer in Mode-B; Vicon Sampled at 250 Hz with MotionMonitor software (Innovative Sports Training, Chicago, IL, USA)	sEMG RMS, Muscle Thickness Change (TC)	14 women
Rohlén et al., 2020 [85]	US	Forearm	9 MHz US Linear Transducer; US Images Sampled at 2000 FPS; 128-Channel DAQ Module	Tissue Doppler, Trigger Pattern, Twitch Train, Twitch Response, and Territory	8 (5 men and 3 women)
Rohlén et al., 2020 [86]	US, iEMG	Arm	9 MHz US Linear Transducer; US Images Sampled at 2000 FPS; 128-Channel DAQ module; Concentric Needle Electrode with 38 × 0.45 mm (AMBU Neuroline, DEN)	Tissue Doppler, US and EMG Trigger Pattern, Twitch Train, Twich Response, and Territory	9 (4 men and 5 women)
Fernandes et al., 2021 [87]	US (WUS)	Forearm	40 mm Linear Probe with 6.6 MHz Center Frequency, B-mode Imaging at 30 FPS. Classifier: LDA	US: DWT-MAV and ENV-LR	5
Zheng et al., 2021 [81]	US, iEMG, sEMG, Reflective Markers, Multi-Electrode Stimulation Matrix	Forearm	5-10 MHz US Transducer; Images Acquired at 54 FPS; iEMG: 0.05 mm Diameter; Reflective Markers Acquired at 100 FPS;	RMS EMG; Flexion Time; US Average Deformation Field and Resulting Field Divergence	2 men

CSA, Cross-Section Area; DAQ, Digital Acquisition Module; DWT, Discrete Wavelet Transform; ENV, Envelope; LR, Linear Regression; MAV, Mean Absolute Value; MPF, Mean Power Frequency; RMS, Root Mean Square; SMMG, Sonomechanomyography; WUS, Wearable Ultrasound.

Each active MU generates a mechanical response, in which the group [85] proposed a method to identify it on a voluntary contraction. The experimental data showed that a MU could be identified with characteristics similar to the MU discriminated by the EMG. This implies that this method can complement the EMG method by adding a 2D spatial resolution and a territorial unit of size to the units simultaneously active in every muscle. The same group [86] also evaluated whether a single MU and its mechanical twitches can be identified using ultrafast US voluntary contraction imaging. The proposal compares the position and firing pattern of the decomposed components of the ultrafast US acquisition with an EMG needle. It was shown for the first time that a single MU and its mechanical twitch could be identified in voluntary contraction experiments. Sequence decomposition of US images provides a non-invasive means to study the neural control of a MU and its mechanical properties.

The effect of US lateral spatial resolution on finger flexion classification was evaluated in [87]. After they analyzed finger flexion performance rating, it was concluded that a US transducer with reduced lateral resolution could perform as accurately as clinical US probes with high resolution under the conditions proposed in the paper.

## 4. Future Trends

The studies presented here in this review point out some approaches that have been investigated in the last decade concerning musculoskeletal assessments with multisensory setups involving ultrasound. Apart from the usual medical trials and diagnosis, most of its applications can be impactful in other fields, such as HMI, wearables, and prosthetic control, among others. Table 3 shows a tradeoff of each technique applied in the HMI field.

Most of the challenges regarding prosthetic control with A-Mode ultrasound are related to system compatibility, transducer shift, and the number of channels involved. As for the B-Mode and M-Mode, there is the need to solve several issues, for example, hardware robustness, high computational costs, and improvement of some digital signal processing (DSP) and filtering techniques. More broadly, related to this matter, research trends point to problems related to ultrasound sensors compatibility, such as wearable ultrasound and miniaturized versions of A-Mode transducers. Nonetheless, B-Mode and M-Mode ultrasound also bring a scope to more detailed applications, mainly when high accuracy is required. 

Many studies [12,27,80] have been applying image tracking software for automatic evaluation of some parameters in ultrasound images to assess patterns in the musculoskeletal system. Furthermore, with the recent improvements in machine learning algorithms [70], a wider and standardized dataset is required to feed and to enhance algorithms prediction and classification patterns. 

In the healthcare field, clinical diagnoses, and physiological studies, the impact of hybrid systems is much more posed to discover specific problems and correlations of techniques in performing specific tasks and activities. However, the more complex and robust the system, the more difficult the setup for dynamic tests becomes, especially with US signal, where the transducer fixation takes a critical role. When the configuration involves multiple signal acquisitions, the entire process requires specific handling. Nevertheless, there is still a necessity to investigate deeper data fusion involving ultrasound to assess musculoskeletal system, as was recently done in NCS, because diagnoses may produce false positive and false negatives, which can be treated with more careful analysis via signal combination.

Therefore, within such limitations and applications in mind, a suggested platform is presented in Figure 5, featuring a multi-sensing system in which the sensors are responsible for capturing the biomedical signal. The acquisition module performs the pre-processing and storage of the captured biosignal data using, for instance, complex field programmable gate array (FPGA)-based systems and/or low-cost open platforms. In this case, an Intel/Altera DE4 Development Board [88] and a Bitalino Board [76] are illustrated. Finally, the digital processing can be performed by high-end research platforms with a central processing unit (CPU)/graphics processing unit (GPU) [21], such as Verasonics Vantage [20], in a programmable environment (such as MATLAB or Python).

## 5. Conclusions

Ultrasound has been a cheap and non-invasive solution for achieving a successful deep analysis at the muscle tissue level, where most of the activity occurs. However, this approach always had a drawback in its outdoor applications. To solve such liability, other biomedical techniques and signals have been applied to evaluate parameters and data. This approach provides an enhanced pattern recognition performance for prosthetic control areas, robust mathematical models, compact integrated systems at the electronic level, and an accurate source for physiological and clinical assessments. Most of the challenges at the electronic level are related to a lack of structured protocols and stable setups to meet signal noises, synchronization, and device compatibility. Furthermore, it can be addressed as more sensors are being added to smartphones and wearable devices [89]. This whole context emphasizes a critical role for hybrid studies and applications. Further, it helps artificial intelligence (AI) technologies such as deep learning to incorporate more datasets for broader research and setups.

## Figures and Tables

**Figure 1 sensors-22-09232-f001:**
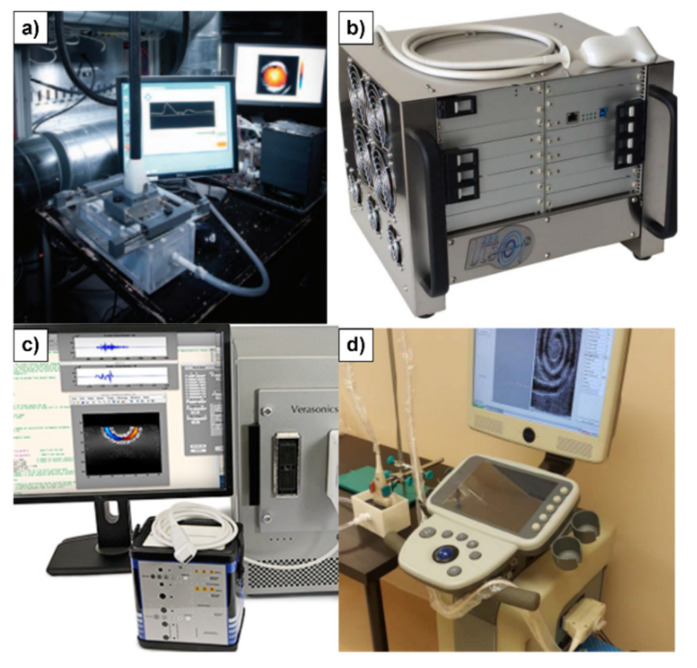
Adapted photographs of Open Architecture Ultrasound Systems: (**a**) SARUS [16], (**b**) ULA-OP [21], (**c**) Verasonics, and (**d**) SonixTouch [21].

**Figure 2 sensors-22-09232-f002:**
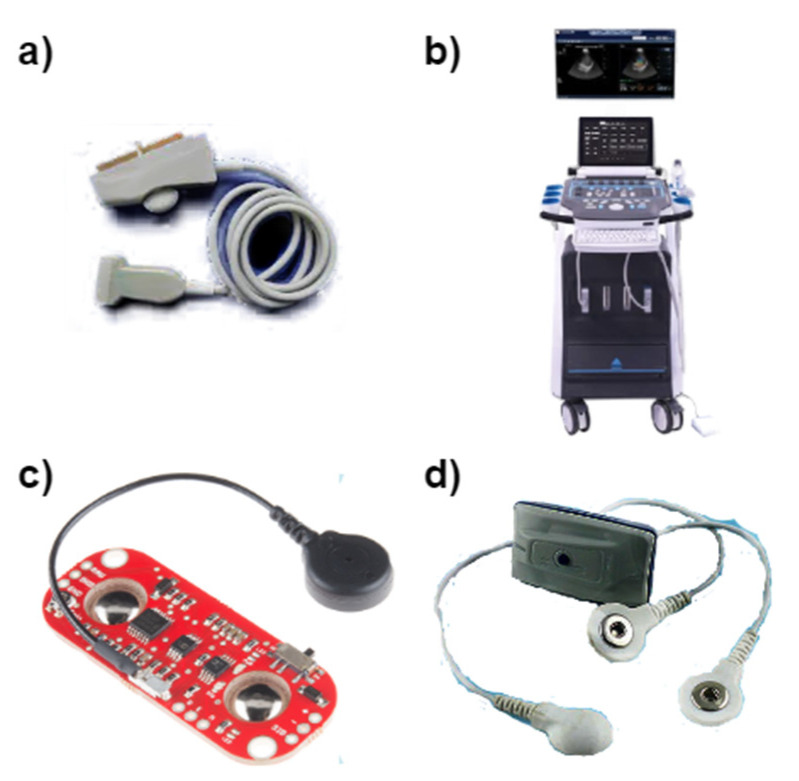
Photographs of biosensing devices (**a**) ultrasound probe for sonomyography, (**b**) ultrasound system for elastography, (**c**) surface electromyography sensor, and (**d**) mechanomyography sensor.

**Figure 3 sensors-22-09232-f003:**
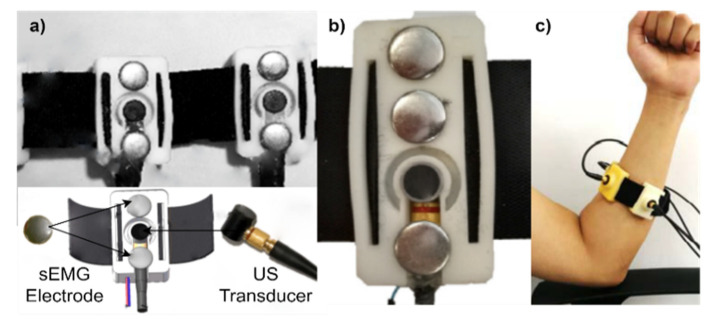
Adapted photographs of wearable systems using ultrasound and sEMG: (**a**) hybrid armband supporting sEMG electrodes and US transducer, developed by authors [62]; (**b**) sensory module integrating US transducer in the black area and sEMG electrodes within gray area, where the reference electrode is on the top, designed by [63]; and (**c**) four-channel A-Mode ultrasound armband, adapted by the group [56].

**Figure 5 sensors-22-09232-f005:**
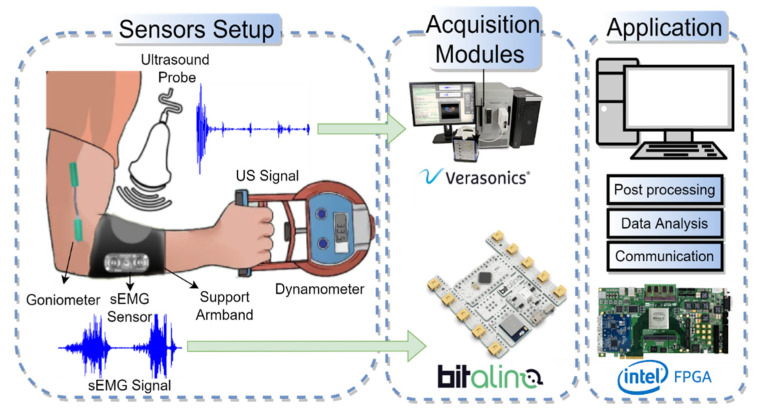
Multi-sensing model for data acquisition with Verasonics and Bitalino interfaces, based on sEMG and ultrasound.

**Table 3 sensors-22-09232-t003:** Biosensing signals for HMI applications.

Signals	Robustness	Advantage	Disadvantage	Filters
A-Mode		Can be Miniaturized, Accuracy	Fixed Position	DAS, Butterworth, Band-Pass, Speckle Reduction
B-Mode	[X]	Muscle Depth, Multiple Directions and High Accuracy	Computational Cost, Size	
M-Mode	[X]	Temporal Resolution, Framed Analysis, and High Accuracy	Extremely High Computational Cost, Size	
sEMG		Excellent Movement Predictor	Noise	Amplifier, Butterworth, Low/High/Band-Pass, Notch, Moving Average
MMG		Can Relate to Muscle Strength Level	Acoustic Interference	Amplifier, Butterworth, High/Low/Band-Pass, Moving Average

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
