# Peer review of "Multi-Sensing Techniques with Ultrasound for Musculoskeletal Assessment: A Review"

_sensors, 2022, doi:10.3390/s22239232_

Round 1
Reviewer 1 Report
A very informative paper. Please address my following comments.
- Please demonstrate the novelty of this review paper in relation to other thematically similar review papers (in the introduction section).
- Please expand your conclusions to include a discussion within the quantitative evaluation of the research.
- No proper future recommendations are given as a way forward for researchers working in this area. This limits the efficacy of this review.
- The overall write-up needs major revision. There is no flow between the paragraphs at many points in the manuscript.
Author Response
Thanks for all the comments, every single suggestion and insight was implemented along the text. We hope the Reviewer find it useful.
- Please demonstrate the novelty of this review paper in relation to other thematically similar review papers (in the introduction section).
In Introduction chapter in the [line 80], was added a text showing the novelties and comparing this work with other review papers along this research field.
- Please expand your conclusions to include a discussion within the quantitative evaluation of the research.
- This work involves a broad set of applications related to ultrasound in combination with other biomedical signals. So, the authors are presenting the use of multiple hybrid systems and devices, especially US systems together with such additional systems. Therefore, the approach selected was not based on presenting quantitative evaluation of such systems, but to present a broad range of such systems and to present a tendency for the following systems in this research area.
- No proper future recommendations are given as a way forward for researchers working in this area. This limits the efficacy of this review.
Chapter 4 from of the manuscript [line 436] was changed to “Future Trends” attend this issue. Text was modified to bring the reader more insights along this topic.
- The overall write-up needs major revision. There is no flow between the paragraphs at many points in the manuscript.
Manuscript was updated to link some missing points and ideas between paragraphs.
Reviewer 2 Report
.Comment 1
In this introduction, the content of the writing is not very readable, and the logic is not strong. It is recommended to modify carefully.
Comment 2
Please provide a detailed comparison table, including the methods, materials, performance of your work with other studies, comments(advantages/disadvantages) and so on.
Comment 3
More information should be introduced as Applications, challenges and future trends
Comment 4
I recommended to authors to write the conclusion in standard ways. Conclusion should be more precise, showing only main results. The future scope should be mentioned.
Comment 5
In section 2 “Biosensing Techniques” the authors just simply explain and mentioned the figure 1 in this section. I recommend to draw figure as figure 1 to explain basic and recent progress of Biosensing Techniques.
Comment 6
The subtitle of figure 2 and 3 are wrong. The authors should clearly explain what is (a), (b) and (c) rather than just mention the references.
Comment 7
Generally, for writing the review papers, author should provide more figures and details to fully provide and support the tittle of paper rather than just summarize it in 4 figures. I recommend to add more figure and information to support the topic.
Comment 8
Some correlated papers in this research field are suggested to be cited and discussed.
For example:
Sardo, F.R., Rayegani, A., Nazar, A.M., Balaghiinaloo, M., Saberian, M., Mohsan, S.A.H., Alsharif, M.H. and Cho, H.-S. (2022). Recent Progress of Triboelectric Nanogenerators for Biomedical Sensors: From Design to Application. Biosensors, 12(9), p.697. doi:10.3390/bios12090697
Author Response
Thanks for all the comments, every single suggestion and insight was implemented along the text. We hope the Reviewer find it useful.
- In this introduction, the content of the writing is not very readable, and the logic is not strong. It is recommended to modify carefully.
Manuscript was updated to link some missing points and ideas between paragraphs.
- Please provide a detailed comparison table, including the methods, materials, performance of your work with other studies, comments(advantages/disadvantages) and so on.
A brief comparison Table was added to Chapter 4 from of the manuscript [line 441].
- More information should be introduced as Applications, challenges, and future trends
Chapter 4 from of the manuscript [line 436] was changed to “Future Trends” attend this issue. Text was modified to bring the reader more insights along this topic.
- I recommended to authors to write the conclusion in standard ways. Conclusion should be more precise, showing only main results. The future scope should be mentioned.
Conclusion was synthetized in a more compact e objective approach, also linking to future trend, discussed in Chapter 4.
- In section 2 “Biosensing Techniques” the authors just simply explain and mentioned the figure 1 in this section. I recommend to draw figure as figure 1 to explain basic and recent progress of Biosensing Techniques.
A NEW Figure was added to SECTION 2 from of the manuscript [line 125] showing photographs of commercial biosensing devices treated presented in text.
- The subtitle of figure 2 and 3 are wrong. The authors should clearly explain what is (a), (b) and (c) rather than just mention the references.
Figure subtitles were adapted to a more detailed version, not only showing the references but pointing to each specification.
- Generally, for writing the review papers, author should provide more figures and details to fully provide and support the tittle of paper rather than just summarize it in 4 figures. I recommend to add more figure and information to support the topic.
At the moment, we manage to include 1 EXTRA figure. If necessary, we could include at least one or two more figures (later on). Sadly, there was NOT enough time to work on more illustrative figures.
- Some correlated papers in this research field are suggested to be cited and discussed. For example: Sardo, F.R., Rayegani, A., Nazar, A.M., Balaghiinaloo, M., Saberian, M., Mohsan, S.A.H., Alsharif, M.H. and Cho, H.-S. (2022). Recent Progress of Triboelectric Nanogenerators for Biomedical Sensors: From Design to Application. Biosensors, 12(9), p.697. doi:10.3390/bios12090697
From this research field it was discussed following papers:
[10] C. Z. H. Ma, Y. T. Ling, Q. T. K. Shea, L. K. Wang, X. Y. Wang, Y. P. Zheng, "Towards wearable comprehensive capture and analysis of skeletal muscle activity during human locomotion," Sensors, v. 19, n. 1, 195, 2019.
[27] Q. Riaz, A. Vögele, B. Krüger, A. Weber, "One small step for a man: Estimation of gender, age and height from recordings of one step by a single inertial sensor," Sensors 2015, 15, 31999–32019.
[52] Y. Ling, C. Ma, Q. Shea, Y. Zheng, "Sonomechanomyography (SMMG): Mapping of Skeletal Muscle Motion Onset during Contraction Using Ultrafast Ultrasound Imaging and Multiple Motion Sensors," Sensors (Basel). 2020 Sep 26; 2019:5513.
[55] S. Grushko, T. Spurný, M. Černý, "Control Methods for Transradial Prostheses Based on Remnant Muscle Activity and Its Relationship with Proprioceptive Feedback," Sensors 2020, 20, 4883. https://doi.org/10.3390/s20174883.
Round 2
Reviewer 1 Report
Thank you for the revisions